# Punishment insensitivity in humans is due to failures in instrumental contingency learning

**Philip Jean-Richard-dit-Bressel[1†], Jessica C Lee[1†], Shi Xian Liew[1], Gabrielle Weidemann[2], Peter F Lovibond[1], Gavan P McNally[1]***

[1]School of Psychology, UNSW, Sydney, Australia; [2]School of Psychology, Western Sydney University, Sydney, Australia

**Abstract** Punishment maximises the probability of our individual survival by reducing behaviours that cause us harm, and also sustains trust and fairness in groups essential for social cohesion. However, some individuals are more sensitive to punishment than others and these differences in punishment sensitivity have been linked to a variety of decision-making deficits and psychopathologies. The mechanisms for why individuals differ in punishment sensitivity are poorly understood, although recent studies of conditioned punishment in rodents highlight a key role for punishment contingency detection (Jean-Richard-Dit-Bressel et al., 2019). Here, we applied a novel 'Planets and Pirates' conditioned punishment task in humans, allowing us to identify the mechanisms for why individuals differ in their sensitivity to punishment. We show that punishment sensitivity is bimodally distributed in a large sample of normal participants. Sensitive and insensitive individuals equally liked reward and showed similar rates of reward-seeking. They also equally disliked punishment and did not differ in their valuation of cues that signalled punishment. However, sensitive and insensitive individuals differed profoundly in their capacity to detect and learn volitional control over aversive outcomes. Punishment insensitive individuals did not learn the instrumental contingencies, so they could not withhold behaviour that caused punishment and could not generate appropriately selective behaviours to prevent impending punishment. These differences in punishment sensitivity could not be explained by individual differences in behavioural inhibition, impulsivity, or anxiety. This bimodal punishment sensitivity and these deficits in instrumental contingency learning are identical to those dictating punishment sensitivity in non-human animals, suggesting that they are general properties of aversive learning and decision-making.

*For correspondence:
g.mcnally@unsw.edu.au

†These authors contributed equally to this work

**Competing interests:** The authors declare that no competing interests exist.

## Introduction

Punishment learning, which encompasses the capacity to encode the adverse consequences of our behaviour, is fundamental to human behaviour. This learning is central to decision-making, assessment of risk, and underpins our ability to adapt to a changing world. Punishment is also a critical tool to promote behaviour change in others. We use fines, threats, censure, social exclusion, incarceration, and so forth to penalise transgressions of personal, moral, and societal expectations. So, successful punishment learning not only maximises probability of our individual survival by reducing any behaviours that may cause us harm, but it also sustains trust, fairness, and mutually beneficial behaviours essential for group cooperation and social cohesion (*Boyd et al., 2010*; *Fehr and Fischbacher, 2003*; *Henrich et al., 2010*).

However, individuals differ markedly in their sensitivity to punishment (*Carver and White, 1994*; *Jean-Richard-Dit-Bressel et al., 2019*; *Marchant et al., 2018*). Insensitivity to punishment can give rise to problematic behaviours that are highly resistant to change. For example, individuals suffering

from substance use disorders, behavioural addictions, and impulse control disorders typically have reduced sensitivity to the adverse consequences of their behaviours (*American Psychiatric Association, 2013*; *Palminteri et al., 2012*) and this reduced sensitivity is associated with an increase in impulsive or risky behaviour. Antisocial personality disorder in adults and conduct disorder, as well as oppositional defiant disorder, in children can be characterised by impaired punishment processing (*Dadds and Salmon, 2003*) associated with persistence of problematic antisocial behaviours. Conversely, depressive disorders have been associated with increased sensitivity to punishment (*Eshel and Roiser, 2010*), causing sufferers to have excessive and often debilitating responses to their perceived failures.

The cause(s) of individual differences in punishment sensitivity remain(s) poorly understood. The first coherent account of these differences was developed by Gray and colleagues, who proposed that punishment sensitivity is dictated by a self-regulatory system of behavioural inhibition that controls avoidance and anxiety in response to aversive events and their predictors (*Corr, 2004*; *Gray, 1972*; *Gray, 1982*; *Gray and McNaughton, 2000*). Temperamental or trait differences in the operation of this behavioural inhibition system were proposed to cause individual differences in punishment sensitivity (*Corr, 2004*; *Gray, 1972*; *Gray, 1982*; *Gray and McNaughton, 2000*). A closely related possibility is that differences in punishment sensitivity are linked to temperamental differences in aversive valuation. The effectiveness of punishment is linked to the aversiveness of the punisher. More aversive punishers suppress behaviour more effectively than less aversive ones and mildly aversive punishers may actually increase behaviour (*Holz and Azrin, 1961*). Differences in aversive valuation will dictate differences in sensitivity to punishment. A separate possibility is that punishment insensitivity is due to reward dominance. That is, in some individuals, choices and behaviour may be more strongly determined by the rewards they earn rather than the punishers they incur (*Gray, 1972*; *O'Brien and Frick, 1996*; *Robinson and Berridge, 2003*). For these individuals, punishment may have negligible effects in the face of competing rewards. A final possibility, which by no means is mutually exclusive to the preceding explanations, is that punishment sensitivity is due to differences in instrumental learning (*Maier and Jackson, 1979*; *Maier and Seligman, 2016*). Successful punishment requires learning that specific actions have adverse consequences and then withholding those specific actions in the future (*Jean-Richard-Dit-Bressel et al., 2018*). Differences in detecting or encoding the instrumental contingency between antecedent actions and their adverse consequences are likely to cause differences in the extent to which punishment will suppress behaviour.

Despite the central importance of punishment to theories of learning, motivation, and decision-making, it has proved difficult to distinguish between these different possible causes of human punishment sensitivity. One reason for this is that although there are a variety of tasks to studying punishment learning (*Bechara et al., 1997*; *Bechara et al., 2005*; *Frank et al., 2004*; *Pessiglione et al., 2006*), few of these tasks actually isolate potential causes of differences in punishment sensitivity. So, research has often relied on modelling underlying learning parameters or seeking self-reports of behavioural and affective responses to potential or hypothetical punishment (e.g., 'I worry about making mistakes', 'Criticism or scolding hurts me quite a bit') (*Carver and White, 1994*), and then correlating these with behavioural (*Corr et al., 1995*; *Fleshler and Hoffman, 1962*; *Frank et al., 2004*; *Kim-Spoon et al., 2016*) and neural (*Adrián-Ventura et al., 2019*; *Fuentes et al., 2012*; *Hahn et al., 2010*; *Pessiglione et al., 2008*; *Reuter et al., 2004*) outcomes, rather than testing the roles of aversive valuation, reward dominance, and learning in punishment sensitivity directly. Mechanistic behavioural assessment of potential causes of differences in punishment sensitivity requires a different approach.

We recently used a conditioned punishment task in non-human animals that allowed us to overcome these limitations and concurrently study individual differences in aversion insensitivity, reward dominance, and contingency learning as causes of differences in punishment sensitivity (*Jean-Richard-Dit-Bressel et al., 2019*). The key advantages of this task were that multiple, competing explanations of punishment sensitivity could be assessed concurrently and more directly than previous studies, and could be mapped using a single behavioural measure. We found a bimodal distribution of punishment sensitivity and showed that punishment insensitivity is due to a failure of

punishment contingency learning that was unrelated to aversion sensitivity, reward dominance, and Pavlovian fear.

Here, we created a novel computer task based on the task in rodents (*Jean-Richard-Dit-Bressel et al., 2019*; *Killcross et al., 1997*) to identify the determinants of human punishment sensitivity. In this task, participants were initially trained to make two responses (R1 and R2) for reward (points gain). Then, a conditioned punishment contingency was introduced for one of these responses (R1) but not the other (R2). During conditioned punishment, R1 continued to earn reward but it also caused presentations of a conditioned stimulus (CS+) followed by punishment (points loss). R2 earned reward and presentations of a different conditioned stimulus (CS-) but no punishment. Participants also had the ability to actively avoid punishment some of the time, through the activation of a shield which prevented the punishment outcome, even after the CS+ was presented. So, four main contingencies were in effect within this task: the instrumental contingency of reward, which should maintain both responses; the instrumental contingency of punishment, which should bias behaviour away from the punished R1 (i.e., cause punishment suppression); the instrumental active avoidance contingency, which should promote avoidance responses (specifically during the CS+); and the Pavlovian CS+/CS- contingency, which should drive aversive learning to the loss-predicting CS+ but not the CS-. This Pavlovian learning was expected to manifest as CS+ elicited reductions in reward-seeking behaviour (a phenomenon known as Pavlovian or conditioned suppression), as well as increased active avoidance behaviour during CS+.

We assessed learning of these four contingencies via a common behavioural measure (click behaviour), as well as via self-reports of outcome value, instrumental and Pavlovian contingency knowledge. This novel approach allowed us, for the first time, to directly and concurrently assess the roles played by behavioural inhibition, aversion sensitivity, reward sensitivity, and instrumental knowledge in explaining differences in human punishment sensitivity. If human punishment insensitivity is attributable to differences in behavioural inhibition or aversion insensitivity, then insensitive individuals should exhibit attenuated Pavlovian reactions and negative valuations of punishment but possess accurate knowledge about the contingencies in effect. If punishment insensitivity is due to reward dominance, then this should be reflected in higher valuations of rewards and responses that earn them, but otherwise intact aversive valuations and contingency knowledge. Finally, if punishment insensitivity is due to failures in punishment contingency knowledge, then insensitive individuals should show normal outcome valuations, intact Pavlovian aversion and Pavlovian contingency knowledge, but impaired punishment contingency learning.

## Results

### Pre-punishment phase

The 'Planets and Pirates' task involved participants (N = 135, 107 female) making mouse click responses on two continuously presented planets (R1 and R2) to earn points. They received two 3 min blocks of this reward training (*Figure 1A*). Each 3 min block used a continuous real-time (i.e., not discrete trial) structure. Both R1 and R2 were rewarded equally (+100 points, 50% probability). Responses and reward delivery were independently registered and immediate visual feedback for these task elements was provided to participants. Under this schedule, points gain was maximised by high rates of R1 and R2.

All participants readily learned the task and accumulated points. There were no significant differences in responding across the two pre-punishment blocks (Block: F(1,134) = 0.085, p=0.771; Block*Planet: F(1,134) = 0.046, p=0.831) (*Figure 1—figure supplement 1*). Therefore, pre-punishment block data were averaged (Pre) to simplify all further analyses. As expected, response rates for R1 and R2 did not differ significantly (t(134) = 0.872, p=0.385) and no preference between R1 and R2 was detected using a normalised measure of response bias (preference ratio, t(134) = 0.512, p=0.610; *Figure 1C* [Pre]).

### Punishment phase

Participants next received three blocks of punishment training (*Figure 1B*). Reward contingencies remained identical to pre-punishment, but additional conditioned punishment contingencies were introduced. R1 now yielded 6 s on-screen presentations of a spaceship (CS+, 20% probability [1.5 s

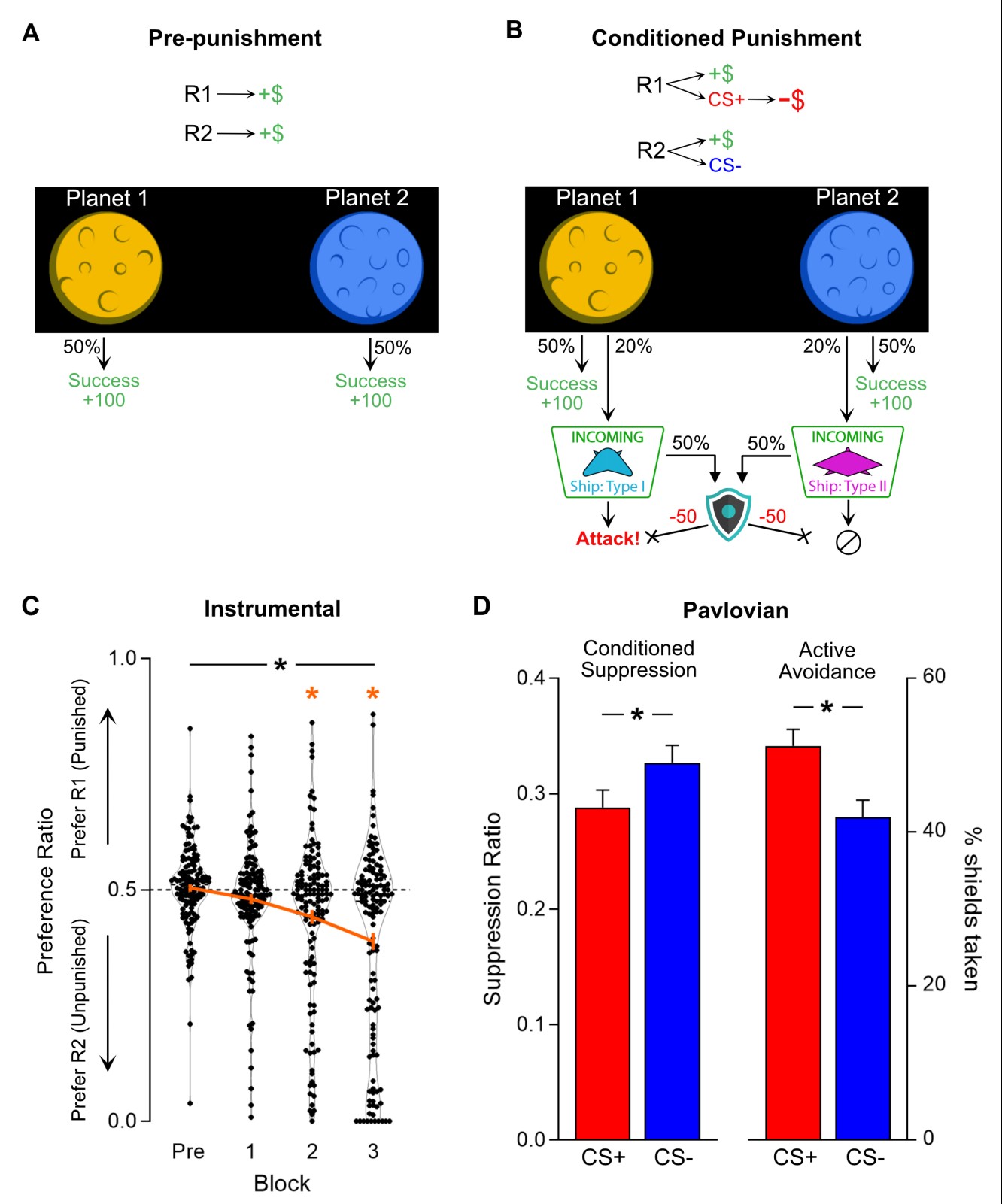

**Figure 1.** Design and aggregate behaviour in 'Planets and Pirates' task. (**A**) During pre-punishment phase, participants could continuously click on two planets (R1 and R2 [side counterbalanced]) to earn reward (+100 points, 50% chance per response). (**B**) During conditioned punishment phase, additional R1→CS+ and R2→CS- contingencies were introduced (20% chance per response). CS+ precipitated attack (−20% point loss), whereas CS- had no aversive consequence. A shield button was made available on a random 50% of CS presentations; activating the shield cost 50 points but

*Figure 1 continued on next page*

*Figure 1 continued*

prevented any point loss from attacks. (C) Preference ratio (orange line = mean ± SEM; dots = individual preference scores) of R1:R2 clicking during pre-punishment phase (Pre) and punishment blocks (1–3). Overall, participants (n = 135) learned to avoid punishment, biasing responding away from punished R1 in favour of unpunished R2. (D) Mean ± SEM CS-elicited behaviour across punishment phase. Participants showed more response suppression (0 = complete suppression) during unshielded portions of CS+ compared CS- (*left panel*), and greater shield use to CS+ than CS- (*right panel*). * [black] p<0.05 behaviour effect; * [orange] p<0.05 vs. null ratio.

The online version of this article includes the following figure supplement(s) for figure 1:

**Figure supplement 1.** Click rate per planet (R1, R2) across pre-punishment blocks.

delay between response and CS presentation]) followed by an 'attack' (−20% of total points), whereas R2 yielded a different spaceship (CS-, 20% probability [1.5 s delay]) and no points loss. During some CSs (random 50%), participants were provided with the opportunity to make an active avoidance response by activating a shield to prevent point loss. If available, the shield button was displayed 3 s after CS onset. Making an active avoidance response by engaging the shield cost 50 points and prevented further reward for the duration of shield presentation (terminating at the same time as the CS). Optimal active avoidance would be to shield whenever possible for CS+ and never for CS-.

Under these conditions, participants as a whole learned the instrumental contingency of conditioned punishment. A preference for the safe R2 over punished R1 developed over blocks (linear trend over blocks: $F(1,134) = 40.97$, p<0.001; *Figure 1C*), with a significant preference ratio for the unpunished R2 during the second ($t(134) = −3.863$, p<0.001) and third punishment blocks ($t(134) = −6.128$, p<0.001).

Participants also learned the Pavlovian contingency between the CS+ and points loss (*Figure 1D*). They exhibited Pavlovian conditioned suppression, reducing rates of responding more during presentations of CS+ than the control CS- ($t(133) = −3.885$, p<0.001; suppression was assessed during unshielded CS relative to CS-free inter-trial interval [ITI]). They were also more likely to actively avoid the CS+ compared to the control CS- by utilising the shield when it was available ($t(129) = 3.199$, p=0.002).

## Individual differences in punishment

So, overall participants learned the instrumental and Pavlovian contingencies in the task. However, there was pronounced variation between participants in this learning. As expected, based on findings in rodents (*Jean-Richard-Dit-Bressel et al., 2019*; *Marchant et al., 2018*), sensitivity to punishment was bimodal. K-means clustering based on final preference ratio identified two clusters (mean silhouette value = 0.74 [greater than three to four cluster solutions]; minimum = 0.04): a smaller punishment-sensitive cluster (n = 43), that finished with a strong preference for unpunished over punished clicking ($t(42) = −24.21$, p<0.001), and an extensive insensitive cluster (n = 92) that did not ($t(91) = 1.902$, p=0.060) (*Figure 2A*). This difference was not pre-existing to the punishment training. Instead, the difference between sensitive and insensitive clusters emerged across punishment blocks (Block*Cluster interaction: $F(1,133) = 223.43$, p<0.001; *Figure 2B*). Sensitive individuals acquired a preference for the unpunished R2 over the punished R1 across blocks ($F(1,42) = 46.55$, p<0.001), whereas insensitive individuals did not ($F(1,91) = 1.111$, p=0.345).

Differences in punishment sensitivity were not simply due to differences in overall responding. Response rates showed that both the sensitive and insensitive participants engaged with similar effort in the task across blocks (Cluster: $F(1,134) = 1.998$, p=0.16; Block*Cluster: $F(1,133) = 0.911$, p=0.342). However, as implied by the preference data, they allocated behaviour differently (Planet*Cluster: $F(1,133) = 58.77$, p<0.001; Planet*Block*Cluster: $F(1,133) = 48.16$, p<0.001). The punishment sensitive cluster reallocated their efforts to the unpunished R2 across blocks (Planet*Block: $F(1,42) = 24.49$, p<0.001), whereas the insensitive cluster did not (Planet*Block: $F(1,91) = 0.364$, p=0.548) (*Figure 2C*). This difference in behavioural allocation was consequential. Although both clusters responded at similar overall response levels, poor punishment learning and maladaptive avoidance significantly reduced total point gain for the insensitive cluster (Cluster: $F(1,133) = 27.81$, p<0.001; *Figure 2D*). Indeed, the insensitive cluster failed to significantly gain points across punishment ($t(91) = 0.223$, p=0.824), whereas the sensitive cluster did ($t(42) = 20.88$, p<0.001).

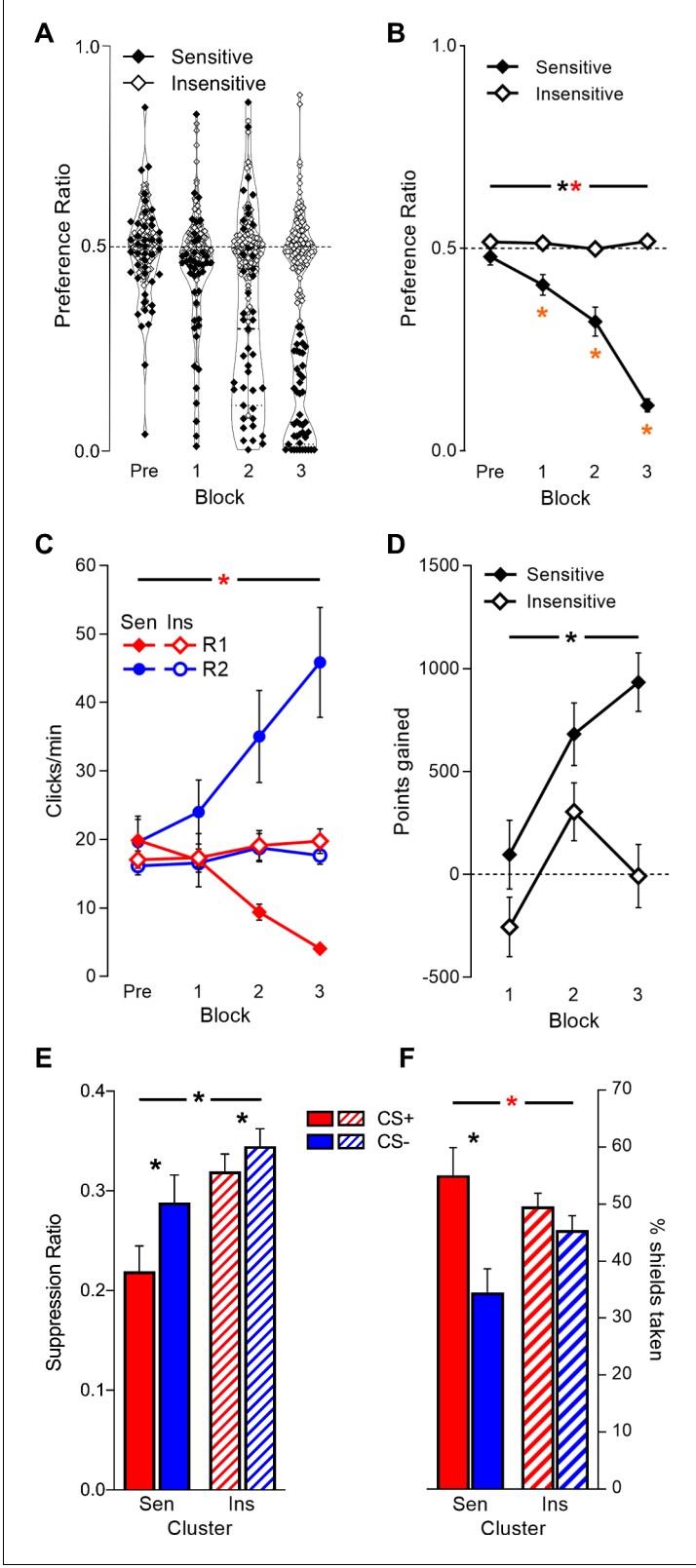

**Figure 2.** Behaviour in task by punishment sensitivity cluster. (A) Final preference ratios (punishment avoidance) were bimodally distributed. Cluster analysis partitioned individuals into punishment-*sensitive* (n = 43; filled dots) and -*insensitive* (n = 92; unfilled dots) clusters. (B) Mean ± SEM preference ratio by cluster across pre-punishment (Pre) and punishment blocks (1–3); the sensitive cluster acquired punishment avoidance, while the insensitive
*Figure 2 continued on next page*

*Figure 2 continued*

cluster did not. (C) Mean ± SEM planet click rates by cluster across pre-punishment and punishment blocks. Clusters exhibited similar overall click rates across task phases, but divergent response allocation. (D) Mean ± SEM point gain per punishment block; only the sensitive cluster achieved a net gain in points across punishment blocks. (E) Mean ± SEM conditioned suppression to CS+ and CS- by cluster. Both clusters showed greater response suppression to CS+ than CS-; sensitive cluster showed greater response suppression overall. (F) Mean ± SEM active avoidance (shield use) by cluster. Only sensitive cluster showed significantly greater shield use during CS + vs. CS-. Sen = sensitive cluster; Ins = insensitive cluster * [black] p<0.05 cluster main effect; * [orange] p<0.05 vs. null ratio; * [red] p<0.05 cluster*behaviour interaction.

Clusters also differed in how they responded to the CSs. The sensitive cluster reduced responding during the CSs significantly more than the insensitive cluster (F(1,132) = 7.221, p=0.008) (*Figure 2E*). This did not significantly interact with CS type but there was a trend to such (Cluster*CS: F(1,132) = 3.649, p=0.058); further analysis revealed both clusters exhibited greater suppression to CS+ than CS- (Sensitive: t(41) = −2.738, p=0.009; Insensitive: t(91) = −2.835, p=0.006). There was, however, a significant interaction between CS type and cluster for active avoidance (F (1,128) = 6.788, p=0.010, *Figure 2F*); the sensitive cluster showed significant discrimination in shield use (t(37) = 3.499, p=0.001) whereas the insensitive cluster did not (t(91) = 1.426, p=0.157). This was despite overall shield use being similar between clusters (F(1,128) = 0.276, p=0.601). So, there was strong evidence for cluster differences in discriminative active avoidance and less so for conditioned suppression.

## Valuation and contingency awareness
### Outcome valuation
The question of interest is what underpins these individual differences in sensitivity to punishment? First, we asked whether the sensitive and insensitive clusters differentially valued reward or punishment (*Figure 3A*). Based on post-block self-report ratings, participants generally liked rewards and disliked attacks. However, in direct contrast to predictions from a reward sensitivity account, the punishment insensitive cluster valued rewards slightly less than the sensitive cluster (F(1,133) = 4.272, p=0.041). Moreover, in contrast to the predictions from an aversion insensitivity explanation, the clusters did not differ in their valuation of point loss (Cluster: F(1,133) = 0.044, p=0.834; Cluster*-Block: F(1,133) = 0.497, p=0.482). So, there was no evidence that differences in punishment sensitivity were due to differences in reward or aversive valuation.

## Pavlovian valuation and contingency awareness
We then asked whether the sensitive and insensitive clusters differed in their self-reported valuation and contingency knowledge of the Pavlovian stimuli (*Figure 3B*). Participants valued the CS+ less than the CS- (CS: F(1,133) = 253.57, p<0.001), with this difference increasing across blocks (CS*Block: F(1,133) = 91.48, p<0.001). Clusters differed in their valuation of the CS+ vs. CS- (CS*Cluster: F(1,133) = 10.41, p=0.002), although this did not interact with block (CS*Cluster*Block: F(1,133) = 1.467, p=0.228). Follow-up analysis showed that this was due specifically to cluster differences in CS- (F(1,133) = 4.727, p=0.031) but not CS+ valuation (F(1,133) = 2.042, p=0.155).

Correspondingly, participants were readily able to correctly attribute attacks to the CS+ as opposed to the CS- (CS: F(1,133) = 220.17, p<0.001, *Figure 3C*), with this contingency knowledge increasing across blocks (CS*Block: F(1,133) = 50.26, p<0.001). This interacted with clusters such that the sensitive cluster exhibited slightly better discrimination between CSs across blocks (CS*Cluster: F(1,133) = 5.273, p=0.023). These cluster differences decreased across blocks (CS*Cluster*Block: F(1,133) = 5.010, p=0.027), with the clusters differing significantly during initial (CS+[1]: F(1,133) = 7.045, p=0.009; CS-[1]: F(1,133) = 6.412, p=0.012) but not later blocks (CS+[2-3]: all F(1,133) ≤ 0.164, p ≥ 0.686; CS-[2-3]: all F(1,133) ≤ 3.920, p ≥ 0.050). Critically, both clusters attributed attacks to CS+ over CS- across each block (Sensitive: all t(42) ≥ 6.74, p<0.001; Insensitive: all t(91) ≥ 4.72, p<0.001).

So, both groups rapidly acquired accurate knowledge about the Pavlovian contingencies. Although punishment sensitive individuals were marginally faster at discriminating attack signalling between the CS+ and CS-, these differences were largely due to differences in the CS- not CS+, and

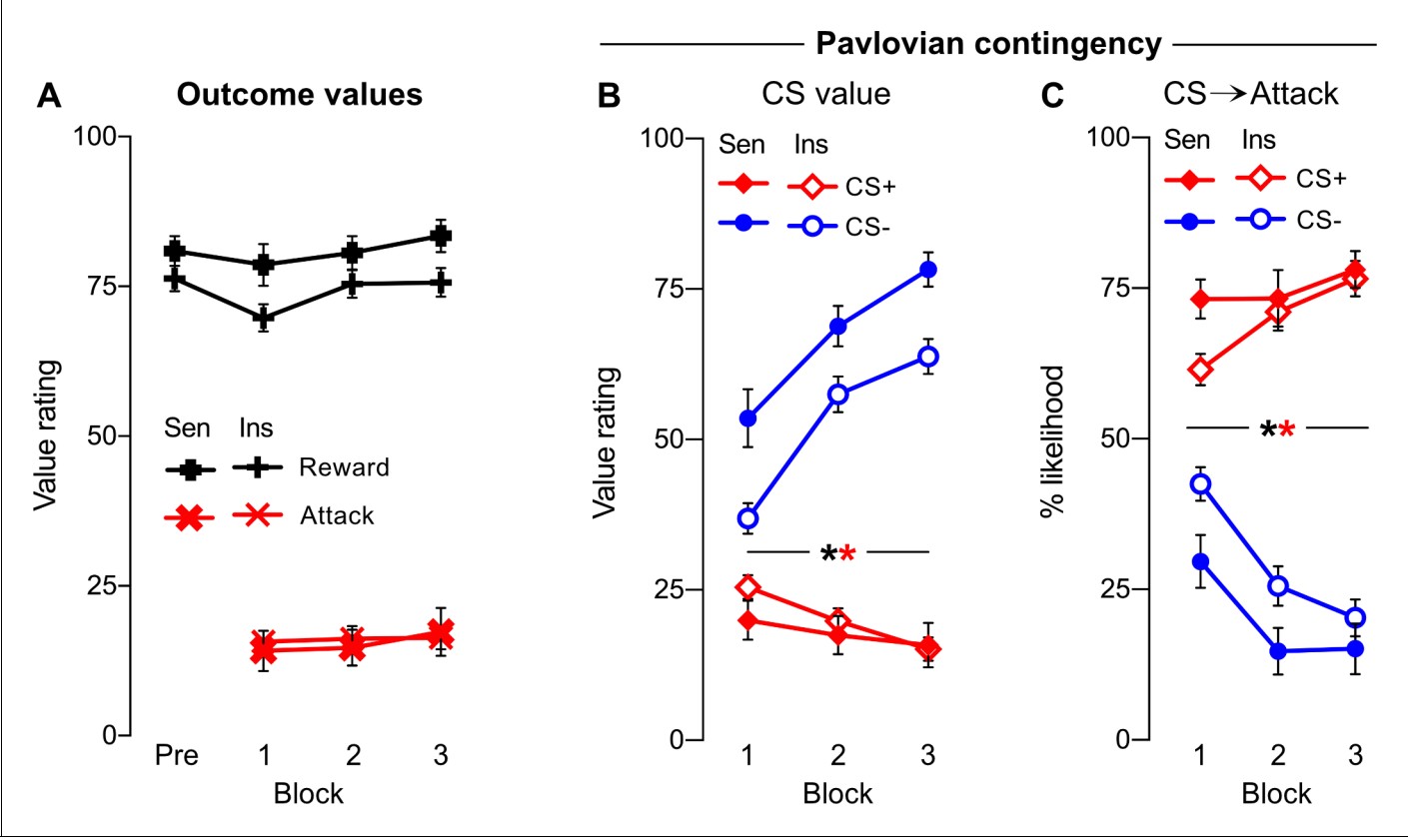

**Figure 3.** Self-reported outcome and conditioned stimulus (CS) valuations, and Pavlovian contingency knowledge. (**A**) Valuation of point outcomes (reward, attack) by cluster across pre-punishment (Pre) and punishment blocks (1–3). Rewards were more highly rated by the sensitive cluster. Both clusters equally disliked attacks. (**B**) Valuation of CS+ and CS- by cluster across punishment blocks. CS+ was valued less than CS-; clusters only differed in their valuation of CS-. (**C**) Pavlovian CS→Attack inferences by cluster across punishment blocks. Attacks were attributed to CS+ over CS-; clusters only differed in attack attributions following first block of punishment. Sen = sensitive cluster; Ins = insensitive cluster * [black] p<0.05 CS main effect; * [red] p<0.05 cluster*CS interaction.

disappeared over blocks. Both clusters showed similarly strong dislike of, and appropriate attack attribution to, the CS+. Thus, insensitive individuals were not greatly impaired in Pavlovian aversive learning, showing intact, stimulus-specific Pavlovian learning about the environmental antecedents of the aversive outcome.

### Instrumental valuation and contingency awareness

We next asked whether the sensitive and insensitive clusters differed in their self-reported valuation and knowledge of the instrumental contingencies. They did. In contrast to the Pavlovian contingencies, but in accordance with their relative profiles of behaviour, sensitive and insensitive clusters differed profoundly in the values they ascribed to their behavioural options. Although both clusters valued the unpunished R2 more than the punished R1, this difference in action value was much greater for the sensitive than insensitive individuals (Response*Cluster: F(1,133) = 42.90, p<0.001, *Figure 4A*). Moreover, this difference in action values increased across blocks (Response*Cluster*Block: F(1,133) = 65.80, p<0.001).

Interestingly, participants acquired subtle yet spurious beliefs about differences in reward probability between responses, estimating a higher reward likelihood for responses on the unpunished planet (Response*Block: F(1,133) = 15.27, p<0.001, *Figure 4B*). However, the two clusters did not significantly differ on this (Response*Block*Cluster: F(1,133) = 3.676, p=0.057), indicating that differential reward attribution was not an obvious source of cluster differences in action valuation or behaviour. The two clusters did differ on how they attributed point loss (*Figure 4C*). Although both

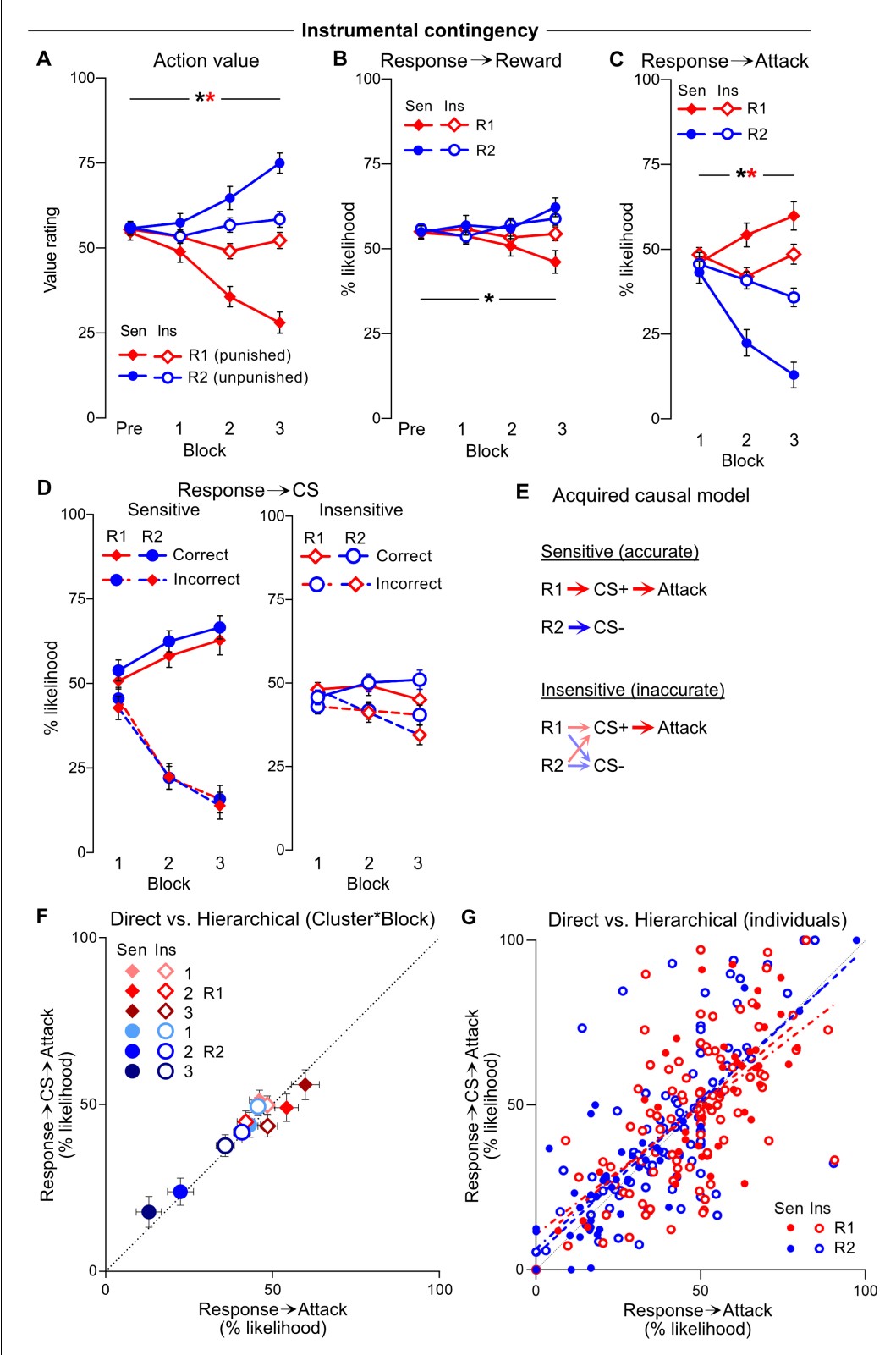

**Figure 4.** Instrumental valuations and contingency knowledge. (**A**) Mean ± SEM valuation of planets (R1, R2) by cluster across pre-punishment (Pre) and punishment blocks (1–3). Unpunished R2 was gradually valued more than punished R1, particularly by sensitive cluster. (**B**) Mean ± SEM instrumental Response→Reward inferences by cluster. Rewards were spuriously attributed to R2 more than R1; this did not interact with cluster. (**C**) Mean ± SEM instrumental Response→Attack inferences. Attacks were attributed to R1 over R2, particularly by sensitive cluster. (**D**) Mean ± SEM instrumental

*Figure 4 continued on next page*

*Figure 4 continued*

Response→CS inferences (*Left panel*: sensitive cluster; *Right panel*: insensitive cluster) according to correct (R1→CS+, R2→CS-) vs. incorrect (R1→CS-, R2→CS+) inferences. Clusters attributed CSs to their respective responses, particularly by sensitive cluster. (**E**) Putative causal model acquired by clusters across punishment phase. Sensitive individuals acquired accurate Response→CS and CS→Attack contingency knowledge. Insensitive individuals acquired accurate CS→Attack knowledge, but failed to acquire accurate Response→CS knowledge. (**F**) Mean ± SEM direct, self-reported Response→Attack inferences vs. estimate computed from hierarchical Response→CS→Attack inferences per response (R1, R2), cluster (Sen, Ins) and punishment block (1–3). Black dotted line represents perfect correspondence between direct and hierarchical inferences. (**G**) Direct, self-reported Response→Attack inferences vs. estimate computed from hierarchical Response→CS→Attack inferences per subject (averaged across punishment). Black dotted line represents perfect correspondence between direct and hierarchical inferences. Dashed line represents lines of best fit for sensitive cluster (per response); dotted-dashed line represents line of best fit line for insensitive cluster (per response). Sen = sensitive cluster; Ins = insensitive cluster * [black] p<0.05 response main effect; * [red] p<0.05 cluster*response interaction.

The online version of this article includes the following figure supplement(s) for figure 4:

**Figure supplement 1.** Relationship between self-reported Response→Attack inferences and estimate computed from hierarchical Response→CS→Attack inferences.

clusters correctly attributed loss to R2 over R1, this instrumental contingency knowledge was considerably greater in sensitive than insensitive individuals (Response*Cluster: $F_{(1,133)}$ = 94.735, p<0.001), and this difference increased across blocks (Response*Cluster*Block: $F_{(1,133)}$ = 23.519, p<0.001).

Response-dependent presentations of the CS+ preceded the attack punisher in this task. We examined Response→CS contingency awareness (*Figure 4D*). Correct inferences (R1→CS+, R2→CS-) and incorrect inferences (R1→CS-, R2→CS+) were assessed. Overall, participants were generally able to learn correct over incorrect associations (Inference: $F_{(1,133)}$ = 129.18, p<0.001), with this discrimination increasing across blocks (Inference*Block: $F_{(1,133)}$ = 54.902, p<0.001), independently of the response (Response: $F_{(1,133)}$ = 0.657, p=0.419). Critically, the insensitive cluster was considerably worse at learning this discrimination (Inference*Block*Cluster: $F_{(1,133)}$ = 27.913, p<0.001), but did not differ in their likelihood to ascribe CSs to responses generally (Cluster: $F_{(1,133)}$ = 0.476, p=0.491). Follow-up analysis revealed that punishment insensitive individuals were less likely to correctly ascribe CSs to their respective responses compared to punishment sensitive individuals (Cluster: $F_{(1,133)}$ = 13.770, p<0.001), while being more likely to ascribe the wrong CS to each response (Cluster: $F_{(1,133)}$ = 21.112, p<0.001).

## Hierarchical conditioned punishment inferences

Both sensitive and insensitive individuals accurately ascribed attacks to CS+, and not CS-. Sensitive individuals also appropriately ascribed CSs to their respective responses, thereby allowing specific ascription of attacks to R1. By contrast, insensitive individuals did not learn the Response→CS contingencies, explaining their poorly discriminated Response→Attack inferences. We determined whether these response and stimulus inferences could be aggregated to map the putative causal models acquired by participants in the task (*Figure 4E*).

To determine whether Response→Attack inferences were mediated by a chain of Response→CS→Attack associations, we computed Response→Attack *predictions* based on the putative mediating inferences (Response→CS and CS→Attack) and compared these with self-reported, direct Response→Attack inferences (*Figure 4F–G*). Across blocks, there was a near one-to-one relationship between direct Response→Attack inferences and those predicted by self-reported mediating inferences for both clusters (*Figure 4F*). This relationship held not only for the Sensitive cluster, which differentiated clearly between R1 and R2, but also for the Insensitive cluster, which largely failed to differentiate between R1 and R2. Response→Attack inferences were effectively predicted via the products of separate Response→CS and CS→Attack inferences (*Figure 4B*) for both R1 (Sensitive: $F_{(1,41)}$ = 35.26, p<0.001, $r^2$ = 0.462; Insensitive: $F_{(1,90)}$ = 53.46, p<0.001, $r^2$ = 0.373) and R2 (Sensitive: $F_{(1,41)}$ = 123.5, p<0.001, $r^2$ = 0.751; Insensitive: $F_{(1,90)}$ = 81.6, p<0.001, $r^2$ = 0.476) (*Figure 4G*). In line with attacks being attributed to CS+ by both clusters (*Figure 3C*), Response→Attack inferences were primarily dependent on Response→CS+→Attack inferences ([first entry in stepwise model] R1: $F_{(1,133)}$ = 75.717, p<0.001, $r^2$ = 0.363; R2: $F_{(1,133)}$ = 103.966, p<0.001, $r^2$ = 0.435), although Response→CS-→Attack also made contributions ([second entry in stepwise model] R1: $F_{(1,132)}$ = +17.483, p<0.001, $r^2$ = +0.075; R2: $F_{(1,132)}$ = +50.449, p<0.001, $r^2$ = +0.155).

Critically, omitting either Response→CS+→Attack or Response→CS-→Attack (particularly CS+) caused a general underprediction of direct Response→Attack inferences (*Figure 4—figure supplement 1*). This suggests both punishment sensitive and insensitive clusters encoded a causal model of the task. Punishment insensitive individuals failed to avoid punishment, not because they could not form or use such models, but instead because they failed to acquire the correct model.

## Relationships between task behaviour and self-report measures

While these findings highlight a likely source of punishment insensitivity in this task, they do not directly link participant behaviour with self-reported valuations and contingency awareness. It is possible that self-report is not an accurate index of the internal representations of value or contingency knowledge that determines behaviour in the task. To address this possibility, we summarised relationships between behaviour, and self-reported valuation and contingency awareness using principal components analysis to assess the underlying correspondences in the data (*Figure 5*).

Variance in instrumental behaviour and self-reported valuation and contingency knowledge was well accounted for by four components (81% of total; *Figure 5A*). Overall click rates across pre-punishment (Pre) and punishment were strongly co-loaded (component 3). These did not co-load with observed response bias (R1:R2 clicking), valuation or inferences. Rather, preference in R1 vs. R2 behaviour co-loaded with instrumental valuation and inferences in each phase. Prior to punishment (Pre), behavioural bias was largely accounted for by bias in valuation and reward inferences (component 2). Response and valuation bias during punishment phase were not strongly related to pre-punishment bias, and instead were inversely related to attack inferences (component 1). Component 4 accounted for variations in spurious reward inference bias, which accounted for some variance in R1: R2 valuation. Together, this shows that instrumental behaviour strongly aligned with explicit contingency awareness.

Stepwise linear regression quantified this alignment between behaviour and inferences. Punishment avoidance (R1:R2 clicking [*Figure 5A*], i.e., click preference ratio) was largely predicted by discriminated Response→Attack inferences (R1:R2→Attack ratio [first entry into model]: $F_{(1,116)}$ = 68.555, p<0.001, $r^2$ = 0.371), with a minor but significant contribution of Response→Reward inferences (R1:R2→Reward ratio [second entry into model]: $F_{(1,115)}$ = +6.191, p=0.014, $r^2$ = +0.032). Correspondingly, using inferences in a stepwise logistic regression model (Response→Outcome ratios per phase), cluster membership was predicted with 84.7% accuracy (Nagelkerke $r^2$ = 0.447) using Response→Attack ratios alone (Response→Reward were not significant predictors).

When comparing Pavlovian behaviour with self-reported valuations and contingency knowledge, three components were sufficient to account for behavioural responses and self-reported valuations and contingency knowledge (69.8% of overall variance; *Figure 5B*). There was strong alignment between CS+:CS- valuation, attack inferences, and active avoidance behaviour (shield use; component 1). That is, the stronger the attribution of attacks to CS+ over CS-, the stronger the bias in value against CS+ and the greater the shield use for CS+ over CS-. This component also predicted higher overall shield use. Surprisingly, neither overall nor discriminated conditioned suppression co-loaded with awareness measures (components 2 and 3). This suggests that Pavlovian contingency awareness more readily predicts instrumental active avoidance than Pavlovian suppression. Indeed, bias in CS→Attack inferences predicted bias in CS+:CS- shield use ($F_{(1,124)}$ = 10.783, p=0.001, $r^2$ = 0.080) but not CS+:CS- suppression.

## Trait measures fail to predict punishment sensitivity

To assess the role that trait characteristics, including behavioural inhibition, aversion sensitivity, impulsivity, and/or reward sensitivity, may play in punishment sensitivity, participants were administered a battery of brief self-report questionnaires at the end of the experiment. This battery included scales for state depression and anxiety (DASS-21) (*Lovibond and Lovibond, 1995*), impulsivity (New Brief BIS-11) (*Morean et al., 2014*), valenced locus of control (Attribution of Responsibility) (*Brewin and Shapiro, 1984*), behavioural inhibition/activation (New Brief BIS/BAS) (*Morean et al., 2014*), and Big five personality traits (Mini-IPIP) (*Donnellan et al., 2006*). Punishment sensitive and insensitive clusters did not significantly differ on any questionnaire subscale (all $F_{(1,133)} \leq 1.794$, p≥0.183), and entry of these subscales into a logistic regression model did not account for cluster membership (Nagelkerke $r^2$ = 0.089).

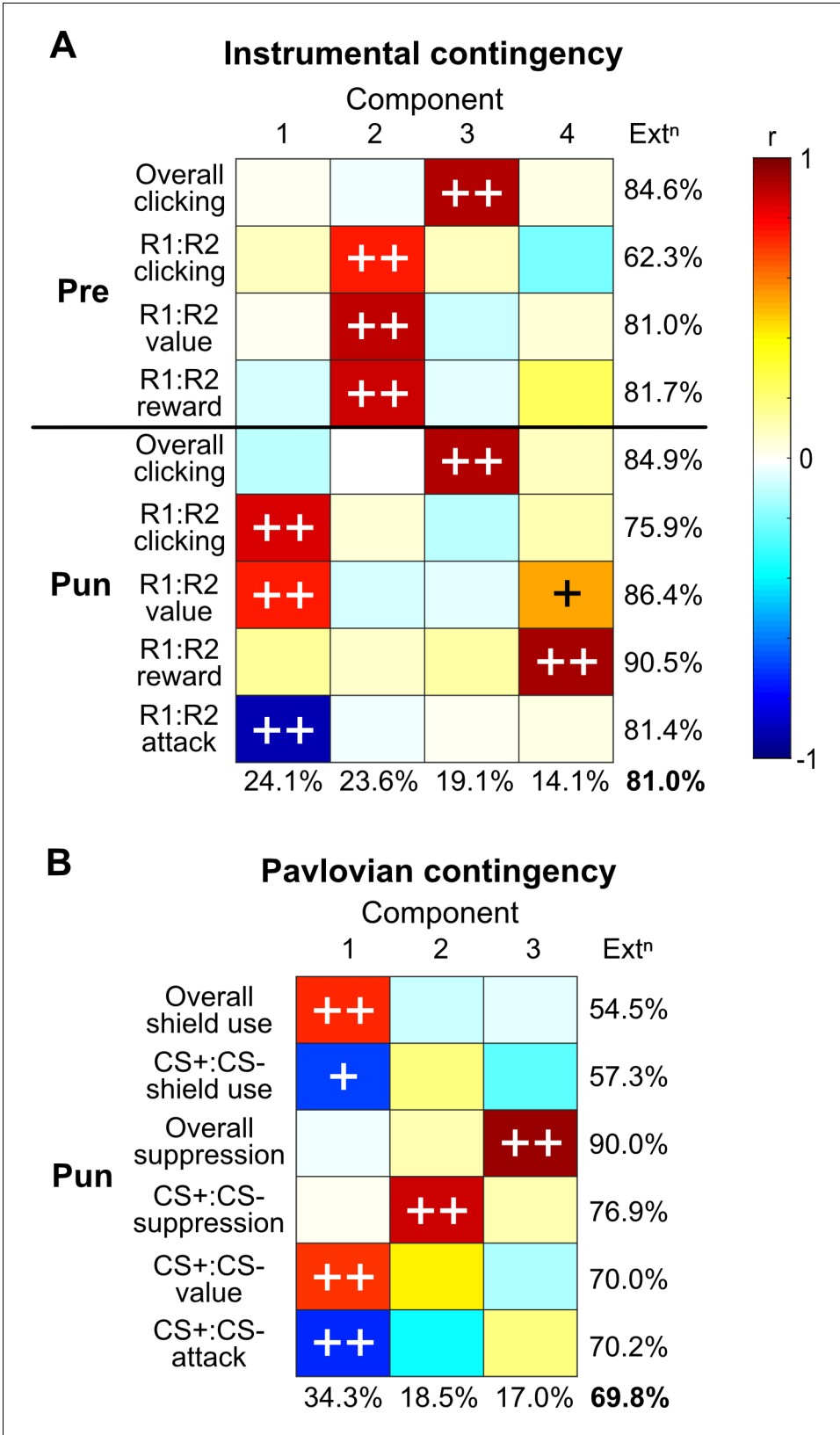

**Figure 5.** Alignments in behaviour, valuations, and contingency knowledge. (A) Principal component analysis of instrumental behaviour, valuations, and contingency knowledge across pre-punishment (Pre) and punishment (Pun) phases. (B) Principal component analysis of conditioned stimulus (CS)-related (Pavlovian) behaviour, valuations,

*Figure 5 continued*

and contingency knowledge across punishment (Pun) phase. Ext$^n$ = overall extraction; ++ = >0.707 loading (>50% variance accounted for by component); + = >0.5 loading (>25% variance accounted for by component).

## Discussion

Punishment involves learning about the adverse consequences of specific actions. Sensitivity to punishment is a fundamental component of adaptive behaviour. However, individuals vary profoundly in their propensity to avoid punishment. Whether this variation is due to differences in motivation, learning, and/or behavioural control has been poorly understood. Here, we used a novel conditioned punishment task to concurrently assess different possible mechanisms underlying punishment insensitivity. Our 'Planets and Pirates' task involved participants making two responses that were equally and independently rewarded, but differentially punished. One response was punished, occasionally yielding a ship cue for point loss (CS+). The other response was unpunished, only yielding a different, safe ship cue (CS-). When assessed as a group, participants learned to avoid the punished response in favour of the unpunished response, while also showing discriminated Pavlovian reactions to CSs. However, these group level trends obscured key individual differences in the data. Punishment avoidance was bimodal, with one cluster showing strong avoidance and the other cluster showing none. These findings are remarkably similar to findings of bimodal punishment sensitivity in non-human animals (*Jean-Richard-Dit-Bressel et al., 2019*; *Marchant et al., 2018*). The presence of bimodality across species suggests that punishment insensitivity is a general feature of normal, mammalian aversive learning and decision-making.

Here, we show that a primary cause of punishment insensitivity was a failure to learn one's instrumental control over aversive outcomes. Punishment insensitive individuals failed to learn that their actions caused the appearance of the cue that signalled punishment, as indicated by their self-reported contingency knowledge. Consequently, they failed to learn the instrumental contingency between their action and that aversive outcome. Punishment sensitivity was defined behaviourally whereas instrumental contingency knowledge and response valuation were assessed via self-report during the task. It is worth noting that behaviour, contingency knowledge, and response valuation were highly consistent with one another. This high level of agreement suggests that the behavioural and self-report measures tapped into the same underlying mechanism of contingency learning. Such a pattern is precisely that expected based on past work showing that learned human behaviour is aligned with self-report measures of contingency awareness (*Lovibond et al., 2011*; *Weidemann et al., 2013*; *Weidemann et al., 2016*).

The failure to learn the relationships between actions and cues, and hence the aversive outcome, may be due in part to the imperfect contingency and delay between the action and CS (*Boakes and Costa, 2014*; *Frankel, 1975*; *Trenholme and Baron, 1975*). However, such delays and imperfect contingencies are common in real life-use of punishment (*Meindl and Casey, 2012*). This poorer instrumental learning impeded their ability to avoid punishment and resulted in the persistence of behaviour despite negative consequences. Precisely the same instrumental learning deficit was recently found to mediate punishment insensitivity in rodents (*Jean-Richard-Dit-Bressel et al., 2019*). This suggests that failure in punishment contingency detection is a critical mechanism dictating punishment sensitivity across species.

Punishment insensitive individuals also failed to learn behaviours that actively prevented punishment. Whereas punishment sensitive individuals took appropriate and selective action (shield use) to actively avoid imminent point loss cued by the CS+ but not the CS-, punishment insensitive individuals did not. Instead, punishment insensitive individuals avoided indiscriminately, actively avoiding both the dangerous CS+ and the safe CS-. This failure to employ active avoidance in a discriminative fashion occurred despite punishment insensitive participants learning and articulating the differences in risk between the CS+ and CS-. So, punishment insensitive individuals both did not withhold behaviour that caused punishment and did not generate appropriate behaviours to selectively prevent punishment. These dual failures of passive and active avoidance learning, observed at different times within the task (during the inter-stimulus interval vs. during stimulus presentations themselves), show that broad differences in learning control over the aversive consequences of behaviour are features of punishment sensitivity.

Equally importantly, our design allowed us to test and exclude potential alternative explanations of these differences. Punishment insensitivity was not due to differences in value, motivation, or Pavlovian (CS→Outcome) learning, as reflected in self-reported valuation of task elements and contingency knowledge. Sensitive and insensitive individuals reported liking rewards to a similar degree and showed equivalent rates of overall reward-seeking. They also equally disliked punishment and did not differ in their valuation of cues that signalled punishment (CS+, point loss). Punishment sensitive individuals did show overall more conditioned suppression than insensitive ones, possibly indicative of modest differences in Pavlovian learning, but the insensitive group nonetheless acquired accurate Pavlovian knowledge and showed discriminated reactions to the Pavlovian cues. So, the failure of the insensitive cluster to respond to punishment was specific to the instrumental contingency.

Interestingly, punishment sensitivity was not explained by key measures previously hypothesised to predict precisely these differences in sensitivity (e.g., behavioural inhibition scale; *Carver and White, 1994*). Moreover, instrumental preference was not accounted for by any of these measures. These findings were unexpected given the widespread influence of these measures, particularly impulsivity and behavioural inhibition/behavioural activation, in theories of punishment avoidance. It was also surprising given the widespread application of these theories and measures to explaining and assessing core learning and behavioural features of punishment sensitivity in clinical populations. It remains possible, of course, that the contributions of these differences were masked by the role of instrumental contingency awareness in dictating punishment sensitivity. However, the key point is that individual differences in accurately encoding this instrumental knowledge precede any other cause of punishment insensitivity. Temperamental differences in behavioural inhibition, aversion sensitivity, or impulsivity can only affect avoidance if instrumental associations are accurately encoded in the first place. As shown here, many individuals have difficulty with this encoding.

The extent to which differences in punishment sensitivity are specific to punishment, or to the specific type of punishment used in the current experiment, will be of interest to determine. Most obviously, whether the individual differences in learning to control the aversive consequences of behaviour are specific to learning about punishers or whether there are symmetrical differences in learning about rewards that dictate variations in reward sensitivity (*Flagel et al., 2009*; *Flagel et al., 2010*; *Flagel et al., 2007*; *Taubitz et al., 2015*). Likewise, it will be of interest to determine to what extent punishment insensitivity, identified here as differences in learning about the controllability of aversive outcomes, is a stable or trait-like characteristic vs. being task-specific. Finally, to what extent these difficulties in encoding the relationship between actions and aversive outcomes precede or predict punishment insensitivity in clinical disorders will be important to consider. There are some data linking punishment insensitivity in clinical populations to failures of contingency awareness (*Blair et al., 2006*; *Schmauk, 1970*), supporting our findings. However, differences in instrumental contingency awareness as a source of individual differences in punishment sensitivity is rarely assessed in clinical populations and we believe this to be an important topic for future work. The task developed here may prove useful to addressing these questions.

In summary, using a novel conditioned punishment task, we showed that punishment insensitivity in humans shares the same behavioural and learning signatures as punishment insensitivity in other animals. Punishment insensitivity emerges specifically from impaired instrumental aversive contingency detection. Punishment insensitive individuals dislike aversive outcomes and predictors of these outcomes, but are less likely to learn their control over them. Whether and how maladaptive decision-making in clinical populations builds upon these profound individual differences in aversive decision-making will be an important consideration for future work.

## Materials and methods

### Participants

Two-hundred and forty-five psychology students from University of New South Wales (UNSW; n = 161 [118 female, 1 other]) and Western Sydney University (WSU; n = 84 [74 female]) were recruited in exchange for partial course credit. The experiment was approved UNSW Human Research Ethics Advisory Panel C (HREAP-C #3385) and WSU Human Research Ethics Committee (HREC #H12809).

Two criteria were used to exclude participants not appropriately engaging in the study: participants were expected to take between 1 and 30 s to answer each question in post-task checks (averaged per check screen), and participants had to correctly answer two catch questions embedded within questionnaires at the end of the study. A total of 135 participants met both criteria (UNSW n = 96 [76 female, 1 other]; WSU n = 39 [31 female]) and were included in further analyses.

## Apparatus and stimuli

The experiment was programmed using the jsPsych library (*de Leeuw, 2015*) and conducted online via the SONA platform. The experiment was programmed to apply fullscreen mode to the browser window. The experiment code and stimuli can be found at https://github.com/jessica-c-lee/planets-task/ (*Lee, 2021* copy archived at swh:1:rev:8e791318e4b22729c8d3e15b61a6f0d17fb7fd68) and https://osf.io/ykun2/. The experiment instructions, questionnaires, and screenshots of the task interface and check screens can be found in the Supplementary information.

## Game interface

During game blocks, participants had mouse control of a custom pointer that turned dark when clicking (visual feedback). Two planets (orange, blue [left/right counterbalanced]) were continuously displayed centre-left and centre-right of the screen (*Figure 1*). The identity of the punished and unpunished planets (left/right) was randomised. A green ring appeared around a planet whenever the mouse pointer hovered over it (visual feedback). Trade signal (reward countdown) was displayed directly beneath each planet, while reward outcomes were displayed directly above each planet. Accumulated points were continuously displayed top-centre of the screen. 'Incoming ship' icons (Type I [turquoise], Type II [purple]; *Figure 1*) were presented in the upper-middle part of the screen. A countdown timer to ship 'encounter' was co-presented immediately below the ship icon. Ship outcomes (attack, attack deflected, nothing) were presented centre-screen, below the encounter countdown. The shield indicator/button was displayed in the lower-middle part of the screen.

## Post-block check screens

For value ratings, icon and descriptor for task elements (planets, ships, outcomes) were each displayed over a slider (0–100). For causal inferences, each antecedent (R1, R2, Ship I, Ship II) received a check screen. The antecedent icon was displayed at the top of the screen, and icons for potential consequences (e.g., ships, outcomes) were displayed over two sliders each (inference [% likelihood], confidence; both 0–100).

## Procedure

At the beginning of the experiment, participants were told that they would be playing a game over several blocks and that their goal was to gain as many points as possible. They were told they could earn points by 'trading' with planets by clicking on them. Following these instructions, they were given a brief multiple-choice comprehension test. Participants had to answer all questions correctly to continue, or else they were returned to the instructions.

## Pre-punishment phase

Pre-punishment phase consisted of two blocks followed by post-block checks. Each game block lasted 3 min (after which 'trading' was suspended, but any remaining cues/outcomes were presented to completion). Responses on either planet (R1 or R2 [left/right counterbalanced]) initiated a 2 s trading signal (countdown), which had a 50% probability of resulting in signalled reward ('Success!+100') or non-reward. R1 and R2 countdowns/rewards were independent of each other, such that both planets could be on countdown. Point gain was maximised by continuous, alternating clicking on both planets, maintaining each on countdown to reward as much as possible.

After each block, value and inference checks were conducted. For value checks, participants were asked on a single screen how they felt about reward and planets (0–100 sliders [Very negative – Neutral – Very positive]). For inference checks they were asked to estimate how often interacting with a planet (one screen per planet) would lead to reward (0–100 sliders [Never (0%) – Sometimes – Every time (100%)]) and how confident they were about this estimate (0–100 sliders [Very uncertain – Somewhat uncertain – Somewhat confident – Very confident]). On each check screen, participants

had unlimited time to make their responses and could click on a 'Continue' button at the bottom of the screen once they had made their ratings. The default slider position was set to 50 (the midpoint of the scale) for all check screens.

### Punishment phase

After pre-punishment, participants were given additional instructions warning of local pirates stealing from traders. Participants were informed that their ship has a shield they can activate to prevent theft, but that it will not always be available. They are also reminded the goal is to have as many points as possible. No information about the contingencies between responding and ships, or ships and their outcomes, was provided.

Participants then received three punishment blocks. Like pre-punishment, punishment blocks lasted 3 min (plus allowance for cue/outcome termination) and R1/R2 responses were independently and equally rewarded with 50% probability. In addition to reward contingencies, responses triggered incoming ship icons (CS+, CS- [Type I or II ship, counterbalanced]). R1 exclusively yielded CS +, whereas R2 exclusively yielded CS- (both 20% probability, 1.5 s delay between the response and CS onset). Only one CS could be triggered at a time. CS+ precipitated attacks (6 s following CS + onset, −20% point loss), displayed via an image file with red 'Attack! -$' text. The CS- had no negative consequence, as indicated via the message 'Ship passed by without incident' in green text. During CS presentations, participants could still make R1/R2 responses and earn rewards unless a shield was active (see below).

At CS onset, a shield charging icon appeared; after 3 s the icon either informed the participant that the shield was unavailable or became an ACTIVATE button. If pressed, the button indicated the shield was active and that 50 points had been deducted. An active shield prevented point loss ('attack deflected' feedback), but also prevented further trading for the remaining duration of the ship (not cued).

Following each punishment block, value and inference checks were again conducted. For value checks, participants were asked how they felt about reward, planets, ships, and attack (set order) on a single screen. For inference checks they were asked to estimate how often interacting with each planet (one screen each) would lead to reward, Ship Type I, Ship Type II, and attack (set order), and how often Ship Type I and Ship Type II (one screen each) led to attack.

### Questionnaires

At the end of the experiment, participants were administered a battery of self-report measures. These included measures for state depression and anxiety (DASS-21 subscales) (*Lovibond and Lovibond, 1995*), impulsivity (New Brief BIS-11) (*Morean et al., 2014*), valenced locus of control (Attribution of Responsibility) (*Brewin and Shapiro, 1984*), behavioural inhibition/activation scales (New Brief BIS/BAS) (*Morean et al., 2014*), and Big five personality (Mini-IPIP) (*Donnellan et al., 2006*). Each questionnaire was administered on one screen each (set order). Two catch questions were embedded within Attribution of Responsibility ('Select the left-most option, strongly disagree, for this question') and New Brief BIS/BAS ('Select three, very true for me, for this question') questionnaires.

### Data analysis

Data was extracted and processed in MATLAB using custom scripts (available at https://github.com/philjrdb/HCP [*Jean-Richard-dit-Bressel, 2021*; copy archived at swh:1:rev:6df52a9f08-fe8150b87b53f004c461ea768bd60f] and https://osf.io/ykun2/), and then imported into SPSS 26 for analysis. Participants that did not meet engagement criteria (1–30 s response times for post-block checks, correct catch questions) were excluded from all subsequent analyses (see *Participants*, *Questionnaires*). Given there were no programmed or observed differences between pre-punishment (Pre) blocks, data from these blocks were collapsed for sake of further analysis.

### Task behaviour

Participant behaviour during the 'Planets and Pirates' task was assessed via clicking on punished and unpunished planets (R1 and R2, respectively), as well as the shield button. Differences in behaviour were analysed using contrasts (see Contrast analysis subsection below).

Instrumental behaviour was assessed using click rates (clicks/min) during non-CS periods (ITI). Combined R1 and R2 ITI rates constituted the overall ITI click rate. These were used to calculate a normalised preference ratio: (R1 ITI rate/Overall ITI rate). These ratios range from 0 to 1, indicating the proportion of clicks that were R1. A score of 0.5 indicates 50% of overall ITI planets clicks were R1 (equal rates of R1 and R2; no preference). A score of 0 indicates a complete preference for R2 over R1, whereas one indicates a complete preference for R1 over R2.

Pavlovian behaviour was assessed using suppression ratios (Pavlovian suppression) and shield use (active avoidance) during ships (CSs). Due to the relative scarcity of CSs and available shields per block, measures were calculated using data aggregated across punishment phase blocks. Suppression ratios per CS were calculated using overall planet click rates during unshielded portions of a CS relative to ITI rates: (overall CS [unshielded] rate/overall ITI rate). These scores range from 0 to 1, with 0.5 indicating equal rates of planet clicking during a CS relative to ITI (no suppression), 0 indicating complete suppression of planet clicking during a CS, and scores above 0.5 indicating an increase in planet clicking during a CS relative to ITI. Shield use per CS was calculated as percentage of available shields taken: (number of shield activations/shields available)*100.

## Self-reported valuation and contingency awareness

Valuation of outcomes, planets, and CSs, as well as contingency inferences between these, were assessed via self-report at the end of each block (see Procedure subsection above). Raw value ratings and inferences (% likelihood rating), each ranging from 0 to 100, were analysed using orthogonal contrasts (see Contrast analysis subsection below).

## Contrast analysis

Behaviour and self-report data across blocks were analysed using within-subject and mixed between- × within-subject ANOVAs (orthogonal contrasts). Where applicable, within-subject contrasts were block (linear), response (R1 vs. R2), CS (CS+ vs. CS-), inference (correct vs. incorrect R→CS). Where applicable, cluster was used as a between-subject contrast (sensitive vs. insensitive). Follow-up analysis of cluster differences were analysed using one-way ANOVA.

Significant preference/suppression ratios were determined using one-sample t-tests against the null value of 0.5.

## Clustering

K-means clustering was used to identify clusters of punishment sensitivity using final block preference ratio as input. Silhouette values were obtained for two to four clusters. The two-cluster solution (punishment sensitive vs. insensitive) was optimal, producing the highest mean silhouette value [0.740] and (unanimously positive silhouette values [minimum = 0.04]).

## Hierarchical/chain inferences

To assess the relationship between direct, self-reported Response→Attack inferences and hierarchical Response→CS→Attack inferences, 'chained' attack probability estimates were calculated using self-reported Response→CS and CS→Attack inferences. CS-specific R1→Attack chain estimates were:

R1→CS+→Attack estimate = (R1→CS+ % likelihood) x (CS+→Attack % likelihood)
R1→CS−→Attack estimate = (R1→CS- % likelihood) x (CS-→Attack % likelihood)

The overall R1→Attack chain estimate was a summation of R1→CS→Attack chain estimates (capped at 100%):

R1→Attack estimate = R1→CS+→Attack estimate +R1→CS-→Attack estimate

The same was done for R2→Attack chain estimates. Chain estimates were calculated per block (*Figure 4F*) and across punishment phase (*Figure 4G*). Linear regression was used to compare Response→Attack chain estimates (CS-specific and overall) against direct Response→Attack inferences.

## Principal component analysis

Relationships between behaviour, valuation, and causal inferences were summarised via principal component analysis. Number of components was determined as that needed to extract at least 50% of each item's variance. Components were varimax rotated to improve interpretability.

For instrumental contingencies, items were overall ITI response rates, preference ratio (ITI response bias), R1:R2 value ratio (response valuation bias), R1:R2→Reward ratio (Response→Reward inference bias), and R1:R2→Attack ratio (Response→Attack inference bias) per task phase.

For Pavlovian contingencies, items were overall shield use %, CS+:CS- shield-use ratio (CS shield-use bias), overall CS suppression, CS+:CS- suppression ratio (CS suppression bias), CS+:CS- value ratio (CS valuation bias), CS+:CS-→Attack ratio (CS→Attack inference bias) across punishment phase.

## Stepwise and logistic linear regression

Stepwise linear regression (p-to-enter $\leq 0.05$, p-to-remove $\geq 0.1$) was used to determine which outcome-related inferences were significant predictors of behaviour. For instrumental behaviour, the dependent variable was average preference ratio across punishment phase, with R1:R2→Reward ratio and R1:R2→Attack ratio (across punishment phase) as predictors. For Pavlovian behaviour, separate regressions were run for CS+:CS- suppression ratio and CS+:CS- shield use, with CS+:CS-→ Attack ratio as predictor.

Logistic regression was used to determine whether R1:R2→Reward ratio and/or R1:R2→Attack ratios were predictors of cluster membership. Response→Outcome ratios were entered in a stepwise manner (p-to-enter $\leq 0.05$, p-to-remove $\geq 0.1$).

## Acknowledgements

This work was supported by the Australian Research Council (DE210100292 [JCL], DP190103738 [PFL]; DP190100482 [GPM]).

## Additional information

### Funding

| Funder | Grant reference number | Author |
| --- | --- | --- |
| Australian Research Council | DE210100292 | Jessica C Lee |
| Australian Research Council | DP190103738 | Peter F Lovibond |
| Australian Research Council | DP190100482 | Gavan P McNally |

The funders had no role in study design, data collection and interpretation, or the decision to submit the work for publication.

### Author contributions

Philip Jean-Richard-dit-Bressel, Conceptualization, Resources, Data curation, Software, Formal analysis, Investigation, Visualization, Writing - original draft, Project administration, Writing - review and editing; Jessica C Lee, Conceptualization, Resources, Data curation, Software, Formal analysis, Funding acquisition, Investigation, Writing - original draft, Project administration, Writing - review and editing; Shi Xian Liew, Resources, Software, Writing - review and editing; Gabrielle Weidemann, Conceptualization, Investigation, Writing - original draft, Writing - review and editing; Peter F Lovibond, Conceptualization, Supervision, Funding acquisition, Writing - original draft, Writing - review and editing; Gavan P McNally, Conceptualization, Supervision, Funding acquisition, Visualization, Writing - original draft, Project administration, Writing - review and editing

## Author ORCIDs

Philip Jean-Richard-dit-Bressel (iD) http://orcid.org/0000-0002-0898-8987
Shi Xian Liew (iD) http://orcid.org/0000-0003-0432-1795
Gavan P McNally (iD) https://orcid.org/0000-0001-9061-6463

## Ethics

Human subjects: Informed consent and consent to publish was obtained. The experiment was approved UNSW Human Research Ethics Advisory Panel C (HREAP-C #3385) and WSU Human Research Ethics Committee (HREC #H12809).

## Decision letter and Author response

Decision letter https://doi.org/10.7554/eLife.69594.sa1
Author response https://doi.org/10.7554/eLife.69594.sa2

## Additional files

### Supplementary files
• Transparent reporting form

### Data availability

All data generated or analysed during this study are available at: https://osf.io/ykun2/ and https://github.com/philjrdb/HCP (copy archived at https://archive.softwareheritage.org/swh:1:rev:6df52a9f08fe8150b87b53f004c461ea768bd60f).

The following dataset was generated:

| Author(s) | Year | Dataset title | Dataset URL | Database and Identifier |
|---|---|---|---|---|
| Jean-Richard-dit-Bressel P | 2021 | Data | https://osf.io/n7sdg/ | Open Science Framework, osf.io/n7sdg/ |

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
