## [Decision Letter]

**Acceptance summary:**

This replication in humans of recent work in rats showing that individual sensitivity to conditioned punishment was best related to perception of the instrumental contingencies rather than differences in reward or punishment sensitivity provides insight into an important source of individual variation in how punishment is integrated into decision making, which is both rigorous and translational. The reliable finding of this in two very different tasks and species is impressive since it lends validity to the tasks use in rodent models of various neuropsychiatric disorders that involve changes in this process.

**Decision letter after peer review:**

Thank you for submitting your article "Punishment insensitivity in humans is due to failures in instrumental contingency learning" for consideration by *eLife*. Your article has been reviewed by 2 peer reviewers, including Geoffrey Schoenbaum as the Reviewing Editor and Reviewer #2, and the evaluation has been overseen by Kate Wassum as the Senior Editor.

Essential revisions:

Both reviewers felt that the new work was an excellent extension of the prior study. Below are some suggestions and comments that should be considered in the revision; however the authors are free to respond however they deem appropriate.

1. There is not much detail in the methods about how specific analyses were done, particularly to support claims that subjects had learned Pavlovian or instrumental associations. It might help if some additional operational details were given to remind the reader more in detail where measures came from – for instance, the valuations and other measures of attack probability used to support claims that continencies were not learned are from the surveys are not clear.

2. Another example of this is the suppression ratios. It is not clear when the suppression is happening. The spaceships only show up after a planet is selected right? So why are they continuing to press after that if they understand the task and contingencies? Basically more to help the reader to understand where the various measure are coming from in the results would be helpful.

3. One caveat might be that many of the conclusions regarding the differences between sensitive and insensitive individuals seem to be drawn from self-report surveys. The details in the manuscript were vague in places on this. But assuming this is the case, it may well be that self-report is not the same as internal representation of values and contingencies. The close correspondence to the animal work makes this less of a concern, but some independent non-verbal confirmation of these would be important going forward.

*Reviewer #1:*

The authors set out to devise a behavioral procedure assessing punishment sensitivity in people. The design is based on prior work in rodents that revealed marked diversity in punishment sensitivity. People showed this same diversity, with groups showing sensitivity vs. insensitivity to punishment. The results are a huge translational success. Observing similar variation in people and rodents, with highly similar tasks will facilitate understanding of the behavioral and neurobiological basis of punishment sensitivity.

*Reviewer #2:*

This study is a research advance in which the authors tested punishment sensitivity in humans based on recent work in rats in which they showed that individual sensitivity to conditioned punishment was best related to rats' ability to perceive the instrumental contingencies rather than differences in reward or punishment sensitivity. Here the replicate that basic finding in humans. In the task, people played a video game in which they obtained points for trading with one of two planets on each trial. After initial play, a pirate spaceship was introduced on some trials in association with selection of one planet. The pirate would result in loss of points, though pressing a shield button could mitigate the loss on some trials. Learning was assessed through responses and also a post-testing interview. The results across 135 participants showed that while everyone learned the basic task, there were distinct groups that were sensitive and insensitive to the introduction of the pirate punishment. As was the case in the rat work, insensitivity did not reflect a failure to value reward or the conditioned punishers similarly to sensitive individuals but rather it reflected a failure to assign the continency for the punishment to the appropriate planet selection. These results are novel and important because they provide insight into an important source of individual variation in how punishment is integrated into decision making, which is both rigorous and translational. Though questions exist concerning its generalization, trait-specificity, task dependence and so forth, the demonstration that it is a reliable finding in two very different tasks and species is impressive since it lends validity to the tasks use in rodent models of various neuropsychiatric disorders that involve changes in this process.

That said, there are some criticisms or issues that could be raised. One major caveat might be that many of the conclusions regarding the differences between sensitive and insensitive individuals seem to be drawn from self-report surveys. The details in the manuscript were vague in places on this. But assuming this is the case, it may well be that self-report is not the same as internal representation of values and contingencies. The close correspondence to the animal work makes this less of a concern of course, but some independent non-verbal confirmation of these would be important going forward. Likewise showing test-test reliability, generalization across task of the differences, etc would be interesting.

I also wondered to what extent features of the specific task are necessary to see the differences. For instance, how important is it that the initial training be conducted without punishment in this or the animal version? Could the insensitive individuals actually be paradoxically quicker to learn about the planets, such that they are reluctant to update their credit assignment when new information shows up? This mechanism of the current effect would be dramatically different from the interpretation that insensitive individuals have a deficit in learning about continencies. Similarly how important is it that there be a CS- that is so similar to the CS+? Could the insensitive individuals have a deficit in discriminating or be overly likely to generalize? These points are not meant as criticisms but simply to note that any one study opens as many questions as it addresses perhaps.

---

## [Author Response]

Essential revisions:Both reviewers felt that the new work was an excellent extension of the prior study. Below are some suggestions and comments that should be considered in the revision; however the authors are free to respond however they deem appropriate.1. There is not much detail in the methods about how specific analyses were done, particularly to support claims that subjects had learned Pavlovian or instrumental associations. It might help if some additional operational details were given to remind the reader more in detail where measures came from – for instance, the valuations and other measures of attack probability used to support claims that continencies were not learned are from the surveys are not clear.

Thank you. We have added more details and better signposting within the Methods to help readers see how we analysed behaviour and task-related self-report measures. We also added the signifier “self-reported” at various points within Results and Discussion to remind readers of the nature of measures being used to inform conclusions.

2. Another example of this is the suppression ratios. It is not clear when the suppression is happening. The spaceships only show up after a planet is selected right? So why are they continuing to press after that if they understand the task and contingencies? Basically more to help the reader to understand where the various measure are coming from in the results would be helpful.

Thank you for highlighting the lack of clarity. CS suppression was defined as the change in planet click rates during unshielded portions of a CS relative to CS-free ITI. Given the continuous nature of the task, participants could still click on planets and obtain rewards during CS presentations, so there was still incentive to make R1/R2 responses during unshielded positions of each CS. Relevant text has been added to Methods.

3. One caveat might be that many of the conclusions regarding the differences between sensitive and insensitive individuals seem to be drawn from self-report surveys. The details in the manuscript were vague in places on this. But assuming this is the case, it may well be that self-report is not the same as internal representation of values and contingencies. The close correspondence to the animal work makes this less of a concern, but some independent non-verbal confirmation of these would be important going forward.

This is an important issue. The reviewers are correct in saying our conclusion regarding cluster differences draw heavily from post-block self-report measures. We believe our analyses of inference chains (Figure 4F-G) and the alignments in behaviour vs. self-awareness measures (Figure 5) address these concerns, at least in part. These analyses showed that self-report measures and participant behaviour (putatively driven by representations of value/contingency) are highly consistent with each other. We have interpreted this alignment to mean that the behavioral and self-report mechanisms were tapping into the same underlying mechanism of contingency learning. Expressed differently, if the self-report measures used here did not capture internal value/contingency representations, then it follows that participant behaviour was dissociated from internal representations of value/contingencies, an unlikely proposition based on contemporary understanding of associative learning. It is also worth noting that the alignment we see here between behaviour and verbalizable contingency knowledge is the norm in the broader human learning literature. We have added additional text to Results and Discussion to address and clarify this for readers.